# Impact of the Diet on the Formation of Oxidative Stress and Inflammation Induced by Bacterial Biofilm in the Oral Cavity

**DOI:** 10.3390/ma14061372

**Published:** 2021-03-12

**Authors:** Ilona Rowińska, Adrianna Szyperska-Ślaska, Piotr Zariczny, Robert Pasławski, Karol Kramkowski, Paweł Kowalczyk

**Affiliations:** 1Medical and Social Center for Vocational and Continuing Education in Toruń, St. Jana 1/3, 87-100 Toruń, Poland; elfi.irb@autograf.pl (I.R.); dyrektor@spm.edu.pl (A.S.-Ś.); 2Toruń City Hall, Business Support Center in Toruń, ul. Marii Konopnickiej 13, 87-100 Toruń, Poland; p.zariczny@torun.direct; 3Veterinary Insitute, Nicolaus Copernicus University in Toruń, Poland, Str. Gagarina 7, 87-100 Toruń, Poland; r.paslawski@umk.pl; 4Department of Physical Chemistry, Medical University of Bialystok, Kilińskiego 1Str, 15-089 Bialystok, Poland; kkramk@wp.pl; 5Department of Animal Nutrition, The Kielanowski Institute of Animal Physiology and Nutrition, Polish Academy of Sciences, Instytucka 3, 05-110 Jabłonna, Poland

**Keywords:** bacterial biofilms, inflammation of the oral cavity, periodontitis, gingivitis

## Abstract

The diet is related to the diversity of bacteria in the oral cavity, and the less diverse microbiota of the oral cavity may favor the growth of pathogenic bacteria of all bacterial complexes. Literature data indicate that disturbances in the balance of the bacterial flora of the oral cavity seem to contribute to both oral diseases, including periodontitis, and systemic diseases. If left untreated, periodontitis can damage the gums and alveolar bones. Improper modern eating habits have an impact on the oral microbiome and the gut microbiome, which increase the risk of several chronic diseases, including inflammatory bowel disease, obesity, type 2 diabetes, cardiovascular disease and cancer. The subject of our consideration is the influence of the traditional diet on the formation of oxidative stress and inflammation caused by bacterial biofilm in the oral cavity. Through dental, biomedical and laboratory studies, we wanted to investigate the effect of individual nutrients contained in specific diets on the induction of oxidative stress inducing inflammation of the soft tissues in the oral cavity in the presence of residual supra- and subgingival biofilm. In our research we used different types of diets marked as W, T, B, F and noninvasively collected biological material in the form of bacterial inoculum from volunteers. The analyzed material was grown on complete and selective media against specific strains of all bacterial complexes. Additionally, the zones of growth inhibition were analyzed based on the disc diffusion method. The research was supplemented with dental and periodontological indicators. The research was supplemented by the application of molecular biology methods related to bacterial DNA isolation, PCR reactions and sequencing. Such selected methods constitute an ideal screening test for the analysis of oral bacterial microbiota. The obtained results suggest that certain types of diet can be an effective prophylaxis in the treatment of civilization diseases such as inflammation of the oral cavity along with periodontal tissues and gingival pockets.

## 1. Introduction

The bacterial flora of the mouth epithelium is a very special and diverse environment of microorganisms that live in it permanently or temporarily, with a positive and negative effect on it. These microorganisms can live together in symbiosis or antibiosis by producing special compounds which determine the qualitative and quantitative composition of the oral microflora in a specific environment such as saliva [1]. This contains substances that inhibit the development of undesirable microorganisms (so-called specific defense), like IgA, IgG, IgM. IgA levels are significantly higher than both IgG and IgM levels [2]. In the blood, IgA occurs mainly (80–95%) in monomeric form [1,2], and is secreted onto the mucosa surface, called secretory immunoglobulin A (SIgA), in dimeric (less often trimeric or tetrameric) form. The main role of IgA is immune responses in Langerhans cells and intraepithelial lymphocytes.

The nonspecific defense against pathogenic microorganisms is performed by lactofferin [3,4,5,6], lysozyme, the sialoperoxidase system, and histidine-rich peptides [7]. The change in the pH of the saliva in the oral epithelium in which the bacteria live is not insignificant as they can change from saprophytic to pathogenic forms. The composition of the bacterial flora of the oral epithelium can also be altered by bactericides and bacteriostatic agents, especially antibiotics, cytostatics and steroids [8]. They inhibit the development of biofilms of specific microorganisms like *L. salivarius* [9], indirectly causing the growth of fungi and bacteria resistant to their action like *P. gingivalis* [10]. Such a situation disturbs the homeostasis of epithelium (the biological balance of microorganisms in the environment), leading to caries and dental diseases. Important is also the host-related factor and the additional chemo-physical factors that could interfere with biofilm formation [11,12].

### 1.1. The Physiological Flora of the Oral Cavity

The physiological flora of the oral cavity has evolved in its development over the centuries, creating a very stable endogenous but diverse system of bacteria, fungi, mycoplasmas, protozoa and viruses. The environment of the oral cavity with settled microflora is the so-called closed ecosystem which includes a microenvironment called ecological niches, the so-called niches are: mouth and lips; buccal epithelium; tongue surfaces; supragingival surfaces of teeth, subgingival surfaces of teeth; epithelium of fissures and pockets, and saliva. Bacteria constitute the most abundant group of about 1000 strains, but many species are not yet known [11,12]. Some of them, such as the red or violet complex bacteria, can cause, apart from periodontal disease, also other disease entities such as shingles or sepsis [13,14]. However, despite the relatively large bacterial colonization by pathogenic strains, the oral cavity can be resistant to infections due to the interaction between nonspecific and specific microorganisms and active components contained in saliva [15]. This situation may change when the so-called acquired microflora called exogenous or transient. It happens in situations where there are mechanical injuries in the body (e.g., weakness, dehydration), systemic diseases (in the case of AIDS, cancer), nutritional deficiencies (lack of iron, proteins, vitamins), hormonal disorders (during puberty, pregnancy, diabetes), antibiotics and chemotherapy, but also in childhood and old age when there are physiological changes in the body [16,17,18,19,20].

The formation of biofilm from all bacterial complexes can cause oral diseases [19]. Bacteria from all complexes, including the red complex, effectively block the body’s anti-inflammatory response to the attack of pathogenic microorganisms. It manifests itself in stopping adhesion of interleukin’s activity, which leads to the development of periodontitis, as a civilization disease that affects the majority of adult populations around the world [17]. Infection of periodontal tissues induces inflammation in which the structures that hold the tooth in the socket are destroyed. The teeth begin to move through the exposed pockets and roots of the teeth, which may lead to its falling out. Microbiological studies on this type of processes play a fundamental role [1,8,14,18,19]. The periodontium and the tooth are then occupied by many pathogenic bacteria of the red complex. The dominant periopathogen is *P. gingivalis* [10,11,16,21,22,23]. Each introduction of a new nonbacterial element into the oral cavity environment, starting from the first erupted tooth and ending with complete dentures, contributes to the creation of new environmental conditions [24,25].

### 1.2. The Role of Nutrients in the Colonization of Microorganisms

The order of microbial colonization depends on the availability of nutrients and overcoming the natural limit of nonspecific immunity. The first species of the so-called pioneering, is transmitted by a macroorganism and obtained from the environment [26,27]. Under the new conditions, microorganisms grow rapidly, creating new ecological communities [28,29]. In new ecological communities, an aerobic bacterial flora is formed, which reduces the oxyreduction potential of a given niche, which creates favorable conditions for the development of anaerobic bacteria [30,31]. Thus, a new autogenous succession is formed, which takes place in several stages: the first pioneering microbes appear in the mouth of a newborn baby after 12–18 h. They are aerobic and relatively anaerobic bacteria, the most common are: *S. oralis*, *S. salivarius*, *S. mitis*, with time *S. gordonii* and *S. anginosus* join them. In the period from 1–7 months, the pioneering flora is more and more diverse, especially anaerobic. Gram negative anaerobes appear, such as *Fusobacterium*, *Provotella*, *Veillonella*, sometimes *Capnocytophaga*, *Leptotrichia*, *Campylobacter*, *Eikenella*. There may be from 0 to 7 species in the mouth. The bacterial flora becomes enriched with the eruption of the milk teeth (1–3 years). There are also species of the genus *Acitinomyces*, *Neisseria*, *Lacobacillus*, *Porphyromonas, Rothia*, and *Actinobacillus* [28,32,33,34,35,36]. The oral microflora achieves relative homeostasis when the body reaches adulthood, but when specific conditions arise, such as general conditions (immune disorders, cancer, or post-chemotherapy conditions), opportunistic microorganisms such as *Klebesiella*, *Candida*, *Escherichia*, *Staphylococcus*, *Pseudomonaris* may appear. The term “bacterial complex” in the scientific nomenclature was introduced by Socransky [28]. Socransky divided the pathogens in the biofilm into six main complexes (inducing dental and periodontal diseases in the oral cavity) and adopted the relationship between bacteria as the criterion for the division, and each complex was assigned the appropriate colors—blue, yellow, green, purple, orange and red [28,29,30,31,32,33,34,35,36,37,38,39,40,41,42,43,44,45,46,47,48]. 

The subject of our considerations is the influence of the traditional diet on the formation of oxidative stress and inflammation induced by bacterial biofilm in the oral cavity. Through dental, biomedical and laboratory research, we want to obtain answers to the problems that bother us, posed in the aforementioned research goals.
1to estimate the influence of food on causing inflammation of soft tissues in the oral cavity in the presence of residual biofilm2to estimate the effect of food on the supra- and subgingival biofilm, is it the same or different?3to analyze the types of food in terms of the possibility of slowing down the development of inflammations in the oral cavity or rejecting the above possibility4to estimate which products in the diet cause the greatest oxidation stress in the oral cavity of mammals.

## 2. Materials and Methods

### 2.1. Microorganisms and Media 

The reference bacterial strains of red complex (*P. gingivalis* ATCC 33277, *T. forsythia* ATCC 43037, *T. denticola* 35405 ATCC) were provided from (LGC Standards, Manchester, U.K.) and growth media were used as described in Kucia [42]. For the research were selected women and men (adult patients), nonsmokers, aged 30 to 80 years, in total 20 people. The subject of our research was not to compare the microbiome of smokers and nonsmokers. First of all, we are interested in the influence of a specific diet on the development of inflammation of the periodontal tissues. 

Therefore, we used dental indicators to evaluate the phenomena occurring in the oral cavity. The tests were not intended to distinguish between smokers and nonsmokers and are therefore not included in the table. In our research, we focused on the effects of four types of diets in blocking gum inflammation. Soft and hard bacterial plaque remaining in the oral cavity for more than 3 days contributes to the initiation of inflammation of soft tissues, which over time, after about 3 months, initiates periodontal disease, which is not indifferent to the health of the macroorganism. We took into account the hygienic dental indicators that determine the presence of soft plaque lasting more than 3 days, initiating inflammation of the soft tissues, turning into hard plaque over time. The deposition of tartar as a result of the deposition of unremoved bacterial plaque on the teeth, maintains the inflammation manifested by bleeding gums, swelling and pain. Based on the analyzed indicators, we were able to estimate the state of oral hygiene and its level.

### 2.2. Methodological Description Related to the Four Types of Diet 

The methodology with volunteers was analyzed according to Kucia et al., 2020 [42]. 

Three days before the visit, they were advised:standard oral hygiene for you—do not change your hygiene habitseat meals consisting mainly of products from the recommended diet (diet type F, B, W, T), and at least finish each meal with a product from the recommended dieton the third day of the diet, meeting at the MSCKZiU dental office in Toruńcompleting the documentationcollection of the bacterial inoculum with saliva on plates with an appropriate culture medium (noninvasive collection—without breaking the tissue continuity)—procedure Aperforming hygienic and periodontal measurements (indicators)—procedure Bremoval of tartar—procedure Ccontinuation of the recommended diet for the next 3 days.after next following 3 days, another bacterial inoculum with saliva on plates with an appropriate culture medium and sequencing (noninvasive collection—without damaging the tissue continuity)—procedure D.

Diet F

Traditional meals, containing proteins, carbohydrates—sugars, fats, vegetables

Diet B
Mainly targeted at protein productsYou can eat other foods as well, but end each meal with a sugar-free protein product such as kefir, yoghurt, cheese, etc.

Diet W
Mainly oriented towards vegetablesother foods can also be eaten, but each meal should end with vegetables, such as radish, watercress, kale, broccoli, kohlrabi, etc.

Diet T
Mainly targeted at foods containing Omega-3 fatty acidsyou can also eat other foods, but each meal should be finished with food containing Omega-3 fats, e.g., fish, especially salmon, herring, mackerel, sardines, seafood, sushi, rapeseed oil, linseed, soybean oil, soy products, nuts, almonds, pumpkin seeds.description and symbol of containers:
A.material collected supragingivally before the procedureB.material collected before the procedure from the gingival fissureC.material collected supragingivally 3 days after the procedureD.material collected 3 days after collected material (surgery) from the gingival fissurepan coding:
1st item—ordinal number of the subject from the collective research document2 diet item F or B or W or T3rd position:
supragingival material before the procedure—Amaterial taken from the fissure before the procedure—B3 days after the procedure, supragingival material—C3 days after the treatment, material taken from the fissure—D

The study concerned the effect of a specific diet on the induction of biofilms in periodontal diseases with the voluntary consent of the volunteers participating in the experiment. The following legal regulations, which were also used in the publication by Kucia et al., 2020 [42]. 

### 2.3. Analysis of Bacterial Biofilms 

The analyzed material (a plaque biofilm which was gently taken from the tooth surface with a loop) was analyzed using the methodology described in the publication of Kucia et al., 2020 [42]. After obtaining bacterial biofilms on the growth plates, they were analyzed sequentially with the use of appropriate primers and the obtained results were analyzed in the blast program (2.11.0). 

### 2.4. Statistical Analysis

The Statistica program (version 12, StatSoft, Tulsa, OK, USA) was used for the research. The examined characteristics in different groups are presented as mean values, with standard deviation. Results were analyzed with one-way Analysis of Variance (ANOVA). When the F ratio was significant, the Tukey test was used. The level of statistical significance was analyzed at *p* < 0.05.

## 3. Results 

### 3.1. Research Techniques—Dental Indicators

Over the years, dentistry as a science has developed many indicators defining, among others the state of oral hygiene. Hygiene determines the health of the oral cavity, and deviations from the accepted standards are a harbinger of impending health problems or the determination of already existing pathological conditions. Several of them were selected for a broader look at the studied phenomena. The indicators are presented in Table 1 and the results are summarized in Table 2.

### 3.2. Analysis of Bacterial Biofilms in Search of Periopathogens of Bacterial Complexes

In our study, we analyzed bacterial biofilms in search of periopathogens of bacterial complexes using microbiological methods with the use of reduction culture, sequential analysis by Kucia and by Sanger methods [42,43] as well stomatological indicators of the oral microbiota after a specific diet. The microbiome growth of pathogenic strains were observed on analyzed dishes (Figure 1).

Sequence analysis of bacterial biofilms grown on plates showed mainly red complex bacteria (Table 3). The obtained results show that the influence of the diet can stimulate the development of beneficial microorganisms, including, for example, *L. salivarius*, and it can have a bactericidal effect on pathogenic bacteria.

The results obtained from sequencing using the Sanger method [43] with the right sets of primers for the identification of bacterial biofilms from the superficial (A, B) and subgastral (C and D) surfaces (Figure 1. The values presented in Table 3 show identifications at the level of 99.99–100% of the analyzed species or strains of pathogenic bacteria of the consecutive six complexes (Table 3).

### 3.3. Oral Microbiota Collected from Volunteers

The results obtained from sequencing indicated that in each of the analyzed bacterial biofilms collected from each of 20 patients and sown on the plates with the reduction inoculation method, marked as A, B, C and D (Figure 2), we could observe the presence of bacteria from each of the 6 bacterial complexes of which the largest share is the bacteria found in complexes: orange > red > yellow > green ≥ violet > blue where the reaction equilibrium was shifted towards a specific pH.

Bacteria from the six groups were now being recruited. A specific diet regulates the pH of our oral mucosa and changes the niche of bacterial biofilms towards beneficial bacteria and not the pathogenic bacteria that induce tooth decay (Table 3).

## 4. Discussion

The obtained results were a good and very simple and cheap training test which allowed us to estimate to what extent the types of specific diets have an influence on the formation of specific bacterial biofilms on the induction of inflammatory conditions of periodontal disease with specific bacterial biofilms. Based on specific periodontological indicators, the dental community, including doctors, dentists, and hygienists, will be able to estimate with 100 percent certainty when and at what time, based on the data obtained from the indicators, a bacterial biofilm will be formed and what antibiotic therapy should be applied. The methods described in the manuscript significantly shorten the time of detection of disease entities induced by persistent bacterial biofilms belonging to different classes. They also allow the assessment of the actual inflammation of the periodontal tissues. By specifying the type of identifier, the doctor can directly estimate what type of bacteria he is dealing with and what treatment should be administered. The applied diet does not exclude the use of its various supplements, which can potentially contribute to a significant improvement in the soft tissues of the periodontium, as described in our previous work, Salistat SGL03 [42] However, a diet itself rich in macro and microelements and vitamins without a significant contribution of carbohydrates and fats significantly contributes to the quality and improvement of vitality and health condition of the periodontium as well as of the teeth themselves, protecting them against early and rapid loss Based on the obtained results (Table 1, Table 2, Table 3 and Table 4, Figure 2) we can state that a specific diet determines the selection of a specific microbiota of the organism in our mouth and the advantage of beneficial bacteria over pathogens from the 6 analyzed complexes. The obtained results suggest that the substances contained in the dietary supplement, which is Salistat SGL03, may also interact with red complexes of bacteria showing cariogenic activity present in the human oral cavity [11,12,41,44].

Chronic inflammation in the oral cavity may lead to the destruction of the alveolar bone, which may consequently lead to tooth loss. This process may involve direct damage to tissues by the secreted toxins of pathogenic bacteria. Recent literature data indicate the effect of bacterial biofilm on various organs and systems of a person, including the nervous system, contributing to the formation of neurodegenerative diseases [19].

*L. salivarius* included in the dietary supplement can also significantly counteract tooth decay [2] and complex anaerobic bacteria [24,35,37], (Table 2 and Table 3, Figure 2). Currently, antibiotic resistance among pathogenic bacteria is becoming more and more common, leading to super-resistance. The recommendations issued by the American Dental Association indicate that systemic antibacterial drugs should be used with rational dosing to avoid side effects affecting the body [17,23]. In addition to vaccines based on Covid-19, the search for new methods enabling faster detection of bacterial biofilms and correlation with the function of a specific diet will allow appropriately targeted clinical and diagnostic tests to be conducted.

Our research has shown that the effects of 4 different types of diets differ depending on the sensitivity of the host [11,12], (Table 2 and Table 3, Figure 2). In fact, the levels of the analyzed pathogens obtained by Sanger sequencing [43] of bacteria (see Table 3), were kept at a constant level, perhaps through the use of energy sources from the diet needed for metabolism [7,23,32,35,36,37,38,39,40,41,42,43,44,45,46]. 

The inhibition of growth by bacteria of the fifth complex in oral colonization is critical because these bacteria may indirectly influence neoplastic diseases related to pancreatic cancer and cardiovascular diseases [43,44,45,46,47,48]. Therefore, the use of a safe traditional diet gives a better effect than any type of food additives or spoilers for determining the effect of pathogenic microorganisms in the diet function and periodontal inflammation [38,39,40,41,42,43,44,45,46].

### Analysis of Dental Indices in a Group of 20 Volunteers in Terms of the Effect of Diet on Soft Tissues in the Oral Cavity 

Over the years, dentistry, as a science, has developed many indicators that determine the state of oral hygiene and health (Table 2 and Table 3, Figure 2). Hygiene determines the health of the oral cavity, and deviations from the accepted norms are a harbinger of upcoming health problems, or the determination of already existing pathological conditions. Twenty volunteers took part in the study and committed themselves to an appropriate diet for the duration of the study. The research group was divided into four subgroups of five people, each of the subgroups was assigned a diet (protein, vegetable, fat—Omega-3 fatty acids and the so-called fast food). After performing the indicators in the subjects after three days of dieting, with standard oral hygiene of each of the subjects, with residual bacterial plaque and tartar, the qualitative research was characterized by the fact that information/unexpected results appeared during the research. After analyzing the indicators, it turned out (it was an unintended effect) that oral hygiene after averaging was at a similar level in all participants. Fuchsin staining and OHI and Pl.I hygiene indicators in the four groups showed similar values in terms of oral hygiene. Hard and soft plaque was found in all subjects. Because each of the studied groups of patients with plaque and tartar followed a specific type of diet, after averaging the results from each diet group, it was found that the patients had a similar oral hygiene status. Regardless of the diet used, the measurements of hygiene indicators were similar. Differences began to appear at the level of plaque build-up in the interdental spaces. The inflammation recognition index was lower in the W and T diets. Fuchsin stains tartar, soft bacterial plaque and plaque (in the form of deposits) resulting from drinking colored drinks. The next two indicators determined the presence and amount of hard and soft plaque. The first index was higher because it was associated with discoloration of tartar, soft bacterial plaque and deposits from consuming colored drinks—hygiene was worse (at the right level). The next two measurement indicators indicated the presence of tartar and soft deposits (bacterial plaque) made with a dental probe (periodontometer) and were at a good level in accordance with the indicator evaluation criteria. According to them, measurements, calculations and results were entered into the test evaluation criteria. Three hygiene indicators were listed here (see Table 2 and Figure 2).

The residual bacterial plaque and tartar determine the formation of gingivitis and maintain this condition. In such situations, chronic inflammation most often occurs, which the person is usually unaware of. The respondents had several dental indicators describing the state of oral hygiene (magenta, Pl.I, OHI, API), as well as the indicator describing one of the symptoms of inflammation (gingival bleeding)—PBI and the periodontal treatment need indicator—CPITN. Indicators: magenta, Pl.I, OHI were used in the shortened option (six teeth were examined), but not the same. The results of individual groups were averaged. When analyzing the results, the following was found [38,39,40]:1The magenta index visually determines the presence of plaque deposits: soft, hard and plaque. After averaging the results in each of the studied subgroups, the results were in the range between 2–3, which indicated a sufficient level of hygiene in each of the studied groups.2The Pl.I index indicates the thickness of the bacterial plaque. After averaging the results in each of the studied subgroups, the results ranged between 1 and 2, which indicated good hygiene in each of the studied groups.3The OHI index determines the presence of soft and hard bacterial plaque (tartar). After averaging the results in each of the studied subgroups, the results ranged between 1 and 2, which indicated good hygiene in each of the studied groups.4The API index determines the presence of bacterial plaque in the interdental spaces, in places that are not easy to clean and favor its retention. When the results were averaged in the subgroups with diet B, T, W, the results were in the range of 53.38–69.72%, which indicated average hygiene. In the subgroup with diet F, the mean result was 82.42%, which meant poor hygiene.5PBI determines the bleeding of the gums, which is one of the symptoms of inflammation of soft tissues. After averaging the results, it turned out that with diet W was 0.09%, with diet B it was 4.6% and with diet T it was 5.05%, which indicated clinically healthy periodontitis. With the F diet, this figure was 21.3%, which indicated moderate gingivitis. There was a noticeable difference between the control subgroup with diet F and the subgroups with diets W, B, T.6CPITN index determines the need for specialist periodontal treatment. It should be noted here that in the subgroups with diets B and T there were two people with a need for periodontal treatment, which may have affected the mean result, and in the subgroups with diets W and F, one person each.

The obtained in vitro results indicated that the dietary components T, F, S and W had a strong antibacterial effect on selected pathogenic bacteria responsible for periodontitis. It is important to interpret the results adequately to the PBI index, namely values in the range of 100–50% indicate severe and unlimited gingivitis, values in the range of 50–20% are defined as moderate, and people on the F diet should qualify here for the next range with values of 20–10% describing mild inflammation, a condition requiring improvement in oral hygiene, our respondents did not qualify, despite the fact that the fuchsin indexes indicated sufficient hygiene, and the API index in the interdental spaces as an average, which in the scale of interpretation of results in the index API was in the “bad” range. We remember that inflammation most often/earliest begins in the interdental spaces, so according to this principle, with the obtained results, we should not get the PBI result below 10%, which proves a clinically healthy periodontium. Our subjects who followed a targeted diet, or at least finished each meal with a targeted diet, had a “clinically healthy periodontal” result.

## 5. Conclusions

1The research group, when averaging the results, was at a similar level in terms of oral hygiene.2Differences appeared when determining the API index, where lower parameters were noticed only in the F diet: it can be assumed that the diet is not insignificant when it comes to the deposition of bacterial plaque in the interdental spaces.3Diet influences the inflammation of soft tissues. The most favorable results were seen in people on the W diet, and vegetables are largely vitamins, some of them antioxidants. Similar results were seen in people with diets B and T. However, it should be noted that in the last two subgroups there were two people eligible for periodontal treatment. In the above subgroups, people with sufficient/good hygiene according to the parameters of the indicators had clinically healthy periodontium.4The difference between people on a strictly defined diet and people on the F diet (with a defined PBI index) differed by two degrees in the scale of interpretation of results and the condition was defined as moderate for this subgroup.5Research clearly suggests that the elimination of carbohydrate-based products from food is significant for soft tissues, even with bacterial plaque.6The best results were obtained by patients with diet W, where the average result was less than 0.1%

## Figures and Tables

**Figure 1 materials-14-01372-f001:**
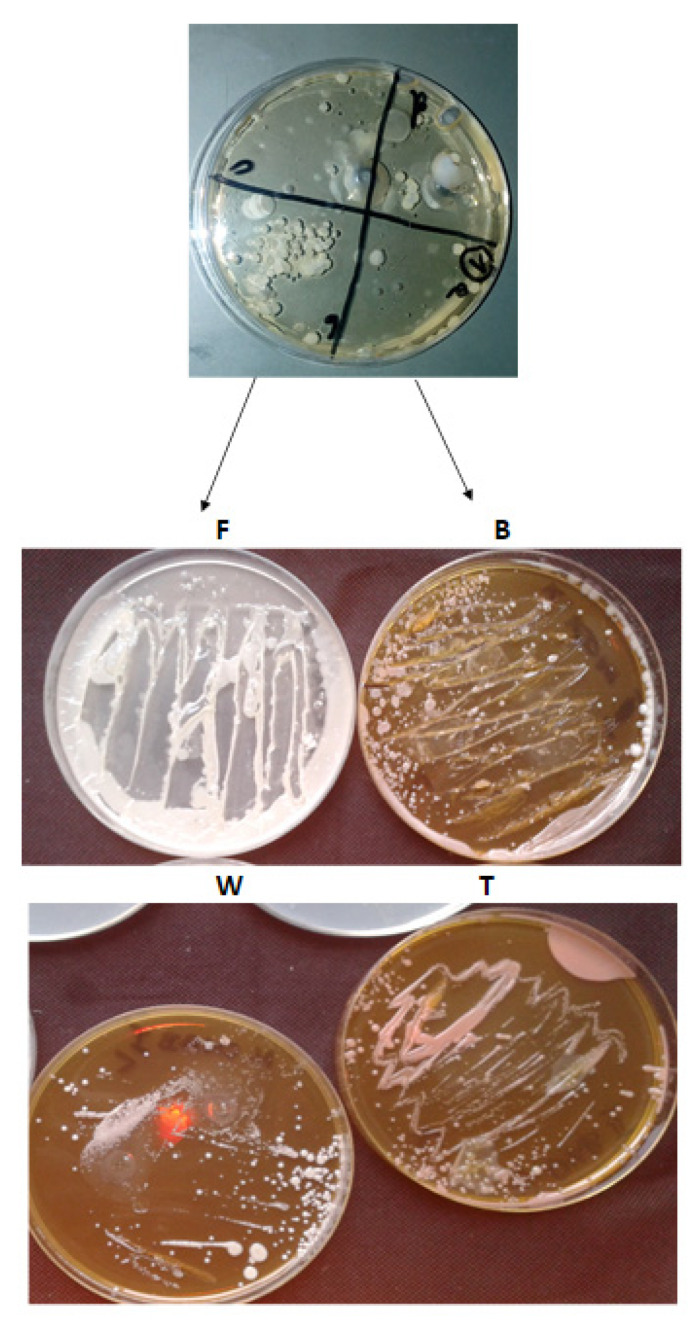
Examples of bacteria on growth media after collecting the bacterial inoculum from patients marked as F, B, W, T (see Section 2.2), relating to each type of diet. Grown fungal colonies (including yeasts) indicate a serious oral cavity infection and insufficient hygiene, therefore only grown bacterial colonies were used for sequencing, in line with the assumptions of the study.

**Figure 2 materials-14-01372-f002:**
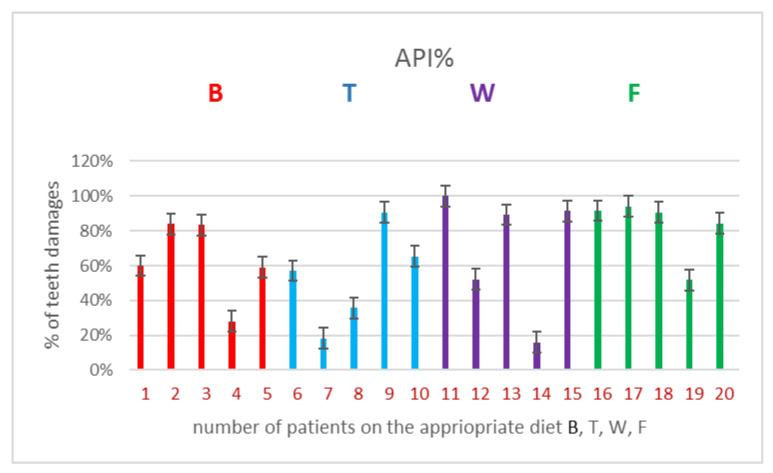
Examples of 4 types of diets used on the selected API indicator (see Table 2. in Results).

**Table 1 materials-14-01372-t001:** The mainly indicators commonly used in periodontology.

Test/Name	What Checks	Examines/Evaluates	Formula—Calculations	Interpretation of the Results
Magenta	stains the bacterial plaque	Entire dentition or 6 teeth (shortened)—one from each group of opposing teeth (16, 14, 11, 36, 34, 31) Incisors, premolars and molars on the outside and inside	F = magenta index X = sum of the buccal values Y = sum of values on the inside n = number of teeth tested F = x + y n	0–2 = good 2–3 = satisfactory 4–6 = bad 0—no sediment 1—the deposit covers the tooth up to 1/3 2—the tooth deposit covers the tooth from 1/3 to 2/3 3—the deposit covers the tooth above 2/3
OHI (Oral Hygiene Index)	Discoloration of dental plaque	16_______11 I_______26 46_______I3 1_______36 upper molars, cheek surfaces lower molars lingual surfaces incisors labial surfaces	OHI = sum of DI and CI values all the examined teeth DI and CI are calculated separately, the 12 results are summed and divided by the 6 teeth Raid/Sediment DI Stone of CI	0–1 = very good hygiene 1–2 = good 2–3 = sufficient over 3—insufficient 0—no tile/stone 1—soft bloom or limescale up to 1/3 2 —soft sediment or limescale up to 2/3 3—soft sediment or scale above 2/3
Pl.I (Plaque Index)	Gingival index of the thickness of the plaque	16___12 I____24_ 44 I___32____36 Four tooth surfaces are examined: inner, outer, distal and mesial	The values of all tooth surfaces are summed up and divided into 4 and further divided into 6 tested teeth Pl.I = sum of area values: 4 Number of tested teeth	0 = healthy, pale pink gums 1 = mild inflammation 2 = moderate inflammation, red gingival margin, pressure bleeding 3 = severe inflammation, red, bleeding gums 0—no tile 1—a thin layer of the plate 2—plaque on observed gingival margin or on the tooth surface and in the fissure 3—abundant accumulation of concrements on the gingival margin and tooth
API (Aproximal Plaque Index)	Examination of the presence of plaque in the interdental spaces (can be dyed)	Interdental spaces in quadrants 1 and 3 from the inside, and in quadrants 2 and 4 from the outside	API = sum of spaces with plate x100 sum of the studied spaces	70–100% = bad 69–40% = average 39–25% = fairly good, within 25% optimal +/−
PBI (SBI) (Papilla Bleding Index)	Gingival—bleeding gums	Interdental spaces in quadrants 1 and 3 from the outside, and in quadrants 2 and 4 from the inside	PBI = sum of bleeding spaces x100 sum of the studied spaces	50–100%= severe and unlimited gingivitis 20–50%= moderate gingivitis 10–20% = mild/condition requiring improvement of oral hygiene Less than 10% = periodontitis clinically healthy +/−
PSI (Parodontaler Screening Index) CPITN (Community Periodontal Index of Treatment Needs)	Indicator of periodontal treatment needs	6 measuring points on each tooth in sextants page. outer proximal, middle, distal page. inner closer, middle, distal Children and adolescents—central incisors and first molars WHO probe	Only the lowest measuring point is recorded ‘measuring 4 in the sextant advances to the next sextant	0 = no further therapy needed 2 = professional removal of subgingival concrements, correction of fillings and prosthetic works 3 and 4 = in any sextant, it is necessary to use periodontal treatment methods after initial therapy +/− 1 = black section of WHO probe fully visible + bleeding,—plaque 2 = black section of WHO probe fully visible + bleeding, + deposits 3 = black section of WHO probe partially visible +/− bleeding, +/– deposits 4 = black section of WHO probe not visible +/– bleeding, +/– deposits

**Table 2 materials-14-01372-t002:** Dental indicators will be used to assess the presence and location of bacterial plaque and tartar in the oral cavity and to determine the needs of specialist treatment—periodontics. The presence of residual bacterial plaque and tartar is a determinant of the presence of inflammation of soft tissues, very often chronic inflammation. The results will be used for further research analysis and for inference.

No	Cod	Magenta Indicator	Indicator. Pl.I Inflammation	Indicator. OHI	Indicator. API	Indicator. PBI	Indicator. PSI/CPITN	Diet	Container. A	Container. B	Container. C	Container. D
1	01.B.(A,B,C,D)	2.2 (good)	1.75 mild	1.1 good	60% average	15% mild, needs improvement	0 (does not require but perio.)	B	5.09.2020	5.09.2020	8.09.2020	8.09.2020
2	02.B.(A,B,C,D)	3 sufficient	1.2 mild	2 sufficient	84% bad	0% healthy periodontium	0 (does not require but perio.)	B	7.09.2020	7.09.2020	10.09.2020	10.09.2020
3	03.F.(A,B,C,D)	2 good	1.25 mild	1.75 good	91.6% bad	66.6% severe gingivitis	3 the patient is eligible for periodontal treatment	F	7.09.2020	7.09.2020	10.09.2020	10.09.2020
4	04.W.(A,B,C,D)	3.3 sufficient	2.66 moderate	3.25 insufficient	100% bad	0% healthy periodontium	0 (does not require but perio.)	W	8.09.2020	8.09.2020	11.09.2020	11.09.2020
5	05.F.(A,B,C,D)	2.5 sufficient	1.69 mild	3.66 insufficient	94% bad	23.53% moderate gingivitis	0 (does not require but perio.)	F	8.09.2020	8.09.2020	11.09.2020	11.09.2020
6	06.T.(A,B,C,D)	2.83 sufficient	1.5 mild	1 good	57.14% average	10.71 condition requiring improvement of hygiene	0 (does not require but perio.)	T	9.09.2020	9.09.2020	12.09.2020	12.09.2020
7	07.T.(A,B,C,D)	1.83 good	0.63 healthy gums	0.5 very good	18.18% improvement of hygiene	0% periodontium wedge. healthy	0 (does not require but perio.)	T	9.09.2020	9.09.2020	12.09.2020	12.09.2020
8	08.B.(A,B,C,D)	1.5 good	1.79 mild	0.9 good	83.33% bad	0% periodontium wedge. healthy	3 the patient is eligible for periodontal treatment	B	10.09.2020	10.09.2020	14.09.2020	14.09.2020
9	09.B.(A,B,C,D)	2 good	1.33 mild	1.83 good	28% good	8% moderate gingivitis	0 (does not require but perio.)	B	10.09.2020	10.09.2020	14.09.2020	14.09.2020
10	10.B.(A,B,C,D)	2.6 sufficient	1.55 good	1.25 good	59.09% average	0% periodontium wedge. healthy	3 the patient is eligible for periodontal treatment	B	12.09.2020	12.09.2020	15.09.2020	15.09.2020
11	11.T.(A,B,C,D)	1.67 good	1.17 mild	1.08 good	35.71% good	10.71% condition requiring improvement of hygiene	4 the patient is needs for periodontal treatment	T	12.09.2020	12.09.2020	15.09.2020	15.09.2020
12	12.T.(A,B,C,D)	3.83 bad	2.08 moderate	1.67 good	90.48% bad	0% periodontium wedge. healthy	3 the patient is eligible for periodontal treatment	T	14.09.2020	14.09.2020	17.09.2020	17.09.2020
13	13.T.(A,B,C,D)	2.5 sufficient	1.83 mild	1.08 good	65.38% average	3.85 % clinically healthy	1 (does not require but perio.)	T	15.09.2020	15.09.2020	18.09.2020	18.09.2020
14	14.W.(A,B,C,D)	3.5 sufficient	1.83 mild	1.25 good	52% average	0% periodontium wedge. healthy	3 the patient is eligible for periodontal treatment	W	15.09.2020	15.09.2020	18.09.2020	18.09.2020
15	15.F.(A,B,C,D)	4 bad	2.38 moderate	1.25 good	84.21% bad	5.26% clinically healthy	2 (does not require but perio.)	F	16.09.2020	16.09.2020	19.09.2020	19.09.2020
16	16.W.(A,B,C,D)	3 sufficient	2 moderate	1.58 good	89.29% bad	0% periodontium wedge. healthy	2 (does not require but perio.)	W	16.09.2020	16.09.2020	19.09.2020	19.09.2020
17	17.W.(A,B,C,D)	2.17 sufficient	1.46 mild	1 very good	16% optimal	0% periodontium wedge. healthy	1 (does not require but perio.)	W	17.09.2020	17.09.2020	20.09.2020	20.09.2020
18	18.W.(A,B,C,D)	2 good	1.63 mild	0.83 very good	91.3% bad	0.43% periodontium wedge. healthy	2 (does not require but perio.)	W	17.09.2020	17.09.2020	20.09.2020	20.09.2020
19	19.F.(A,B,C,D)	2 good	1.79 mild	1.17 good	90.47% bad	0% periodontium wedge. healthy	2 (does not require but perio.)	F	18.09.2020	18.09.2020	21.09.2020	21.09.2020
20	20.F.(A,B,C,D)	3.5 sufficient	2.13 mild	2 good/sufficient	51.85% average	11,11 in need of improved hygiene	2 (does not require but perio.)	F	19.09.2020	19.09.2020	22.09.2020	22.09.2020

**Table 3 materials-14-01372-t003:** Sanger sequencing of bacterial inoculum from specific dishes taken as A, B, C and D.

lp	Type	lp	Type	lp	Type	lp	Type
1a	*Propionibacterium acnes*	1b	*Arachnia propionica (Actinomyces propionicus)*	1c	*Bifidobacterium dentium*	1d	*Eubakterium timidum*
2a	*Streptococcus constellatus*	2b	*Streptococcus oralis*	2c	*Streptococcus mitis*	2d	*Prevotella nigrescens*
3a	*Streptococcus pygenes*	3b	*Streptococcus mutans*	3c	*Streptococcus sanguinis*	3d	*Streptococcus gordonii*
4a	*Capnocytophaga ochracea*	4b	*Capnocytophaga sputigena*	4c	*Capylobacter concisus*	4d	*Capnocytophaga ochracea*
5a	*Rothia dentocariosa*	5a	*Rothia dentocariosa*	5c	*Trichomonas tenax*	5d	*Treponema denticola*
6a	*Lactobacillus acidophilus*	6b	*Lactobacillus buchneri*	6c	*Bifidobacterium dentium*	6d	*Bifidobacterium dentium*
7a	*Lactobacillus casei*	7b	*Lactobacillus plantarum*	7c	*Lactobacillus fermentum*	7d	*Lactobacillus salivarius*
8a	*Streptococcus sanguinis*	8b	*Peptostreptococcus micros*	8c	*Peptostreptococcus micros*	8d	*Streptococcus gordoni*
9a	*E. coli R2*	9b	*Lactobacillus buchneri*	9c	*Bifidobacterium dentium*	9d	*Lactobacillus casei*
10a	*E. coli R4*	10b	*E.coli R4*	10c	*Lactobacillus casei*	10d	*Lactobacillus plantarum*
11a	*E. coli R3*	11b	*Propionibacterium acnes*	11c	*Propionibacterium acnes*	11d	*Desulfococcus oleovorans Hxd3*
12a	*Tanarella forsythia*	12b	*Tanarella forsythia*	12c	*Tanarella forsythia*	12d	*Tanarella forsythia*
13a	*Escherichia. coli R2*	13b	*Escherichia. coli R2*	13c	*Escherichia. coli R2*	13d	*Eubacterium nodatum*
14a	*Pyrococcus sp. OT3*	14b	*Pyrococcus sp. OT3*	14c	*Pyrococcus sp. OT3*	14d	*Pyrococcus sp. OT3*
15a	*Porhyromonas gingivalis*	15b	*Actinomyces odontolyticus*	15c	*Peptostreptococcus micros*	15d	*Peptostreptococcus micros*
16a	*Thermospiro melanesiensis BI429*	16b	*Thermanaerovibrio acidaminovorans DSM 6589*	16c	*Treponema denticola*	16d	*Treponema denticola*
17a	*E. coli R3*	17b	*Lactobacillus buchneri*	17c	*Bifidobacterium dentium*	17d	*Desulfonatronospira thiodismutans Aso3-1*
18a	*Fusobacterium nucleatum*	18b	*Fusobacterium nucleatum*	18c	*Fusobacterium polymorphum*	18d	*Fusobacterium polymorphum*
19a	*Actinomyces odontolyticus*	19b	*Actinomyces* *meyeri*	19c	*Escherichia. coli R2*	19d	*Eubacterium nodarum*
20a	*E* *scherichia* *. coli R3*	20b	*Lactobacillus delbrueckii*	20c	*Bifidobacterium dentium*	20d	*Arachnia propionica*

**Table 4 materials-14-01372-t004:** Statistical analysis of all 20 analyzed patients with different type of diet after application of API I PBI indicator *p* < 0.05 *, <0.01 **, <0.001 ***. (see Section 2.4 in Materials and Methods).

No of Patients/Type of Diet in Indicator	1B	2B	3B	4B	5B	6T	7T	8T	9T	10T	11W	12W	13W	14W	15W	16F	17F	18F	19F	20F
Indicator API	*	*	*		*	*		*	*	*	*	*	*		*	***	***	***	**	***
Indicator PBI	**		*		*	*			*	*	*	*	*	*	*	*	**	**	**	**

## Data Availability

The data presented in this study is available upon request of the respective author.

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
