# Peer review of "Impact of the Diet on the Formation of Oxidative Stress and Inflammation Induced by Bacterial Biofilm in the Oral Cavity"

_materials, 2021, doi:10.3390/ma14061372_

Round 1

Reviewer 1 Report

Dear Authors,

I've really appreciated Your manuscript, it is interesting and well conducted.

I've just some notes to improve it before publication.

In keyword section please be sure to use Medical Subject Headings

In introduction section please create 2 subparagraph (background and aim or similars). unfortunately introduction section is too long and hard to follow, so better state the aim of the manuscript.

In discussion section You could consider to update the text with 2 recent references:

  1. Fiorillo, L. We Do Not Eat Alone: Formation and Maturation of the Oral Microbiota. In Biology, 2020; Vol. 9, p 17.

please specify the host related factor and the additional chemo-physical factors that could interfere with biofilm formation. 

2. Fiorillo, L.; Cervino, G.; Laino, L.; D’Amico, C.; Mauceri, R.; Tozum, T.F.; Gaeta, M.; Cicciù, M. Porphyromonas gingivalis, Periodontal and Systemic Implications: A Systematic Review. In Dentistry Journal, 2019; Vol. 7, p 114.

Conclusion section is too long, please modify it and move some section to the end of discussion.

Author Response

1 recenznet

Open Review

Dear Authors,

I've really appreciated Your manuscript, it is interesting and well conducted.

I've just some notes to improve it before publication.

Thank you very much for your apt suggestions that will improve the substantive value of our manuscript. All corrections in the text are marked in green.

In keyword section please be sure to use Medical Subject Headings

Names have been corrected and completed:

Keywords: bacterial biofilms; inflammation of the oral cavity; periodontitis, gingivitis

In introduction section please create 2 subparagraph (background and aim or similars). unfortunately introduction section is too long and hard to follow, so better state the aim of the manuscript.

In accordance with the reviewer's recommendation, the introduction contains subchapters and a separate subchapter containing the assumptions and aim of the work

1.1. The physiological bacterial flora of the oral cavity

1.2. The role of nutrients in the colonization of microorganisms

2.Aim of the work

In discussion section You could consider to update the text with 2 recent references:

1.Fiorillo, L. We Do Not Eat Alone: Formation and Maturation of the Oral Microbiota. In Biology, 2020; Vol. 9, p 17.

please specify the host related factor and the additional chemo-physical factors that could interfere with biofilm formation. 

The statement was added as a separate sentence before citation 11 and 11a.

  1. Fiorillo, L.; Cervino, G.; Laino, L.; D’Amico, C.; Mauceri, R.; Tozum, T.F.; Gaeta, M.; Cicciù, M. Porphyromonas gingivalis, Periodontal and Systemic Implications: A Systematic Review. In Dentistry Journal, 2019; Vol. 7, p 114.

Listed as 11 Jan 11a 

Conclusion section is too long, please modify it and move some section to the end of discussion.

The summary section has been modified by removing an excerpt from the conclusions for discussion (highlighted in green).

Reviewer 2 Report

In this study, the authors investigated the effect of diet on bacterial biofilm formation. The study reports certain diets could confer a bacteriostatic effect on periodontal pathogens.

The reported experiments and methods are neither well designed and nor aptly described. Overall, the results do not support the stated conclusion and the manuscript needs to be edited for language. The following points also need to be addressed:

  1. Line 38, what do the authors mean by microorganisms can live in antibiotics?
  2. Line 49, what does immune and immune responses mean?
  3. Line 59, Please provide an original article as a reference for this part of introduction “causing 58 the growth of fungi and bacteria resistant to their action like gingivalis”. Reference# 10 is a review paper.
  4. Lines 218 to 223, is this part of the introduction?
  5. How the samples were collected? What were the samples? Was it plaque or gingival crevicular fluid?
  6. Information on the volunteers is missing; age, gender, smoking status…… etc
  7. Line 176, What was the procedure? Please, explain it
  8. Do the authors mean gingival sulcus by the gingival fissure?
  9. Line 179, what was the surgery? Please, explain it
  10. What was the growth condition of inoculated bacteria?
  11. What the white colonies represent in figure 3?
  12. Results of figure 4 should be included in the results
  13. A major part of the discussion is not related to the results and findings

Author Response

In this study, the authors investigated the effect of diet on bacterial biofilm formation. The study reports certain diets could confer a bacteriostatic effect on periodontal pathogens.

The reported experiments and methods are neither well designed and nor aptly described. Overall, the results do not support the stated conclusion and the manuscript needs to be edited for language. The following points also need to be addressed:

 Thank you very much for your apt suggestions that will improve the substantive value of our manuscript. All corrections in the text are marked in green.

  1. Line 38, what do the authors mean by microorganisms can live in antibiotics?

We are very sorry of course this is a translation error, we meant that bacteria can live with each other in symbiosis or antibiosis and not in antibiotics. Corrections are marked in green in the manuscript

  1. Line 49, what does immune and immune responses mean?

We are very sorry of course this is a translation error, the fix was corrected it was about the immune response

  1. Line 59, Please provide an original article as a reference for this part of introduction “causing 58 the growth of fungi and bacteria resistant to their action like gingivalis”. Reference# 10 is a review paper.

New publications have been included in the literature list as 11 and 11a

Fiorillo, L. We Do Not Eat Alone: Formation and Maturation of the Oral Microbiota. In Biology, 2020; Vol. 9, p 17.

Fiorillo, L.; Cervino, G.; Laino, L.; D’Amico, C.; Mauceri, R.; Tozum, T.F.; Gaeta, M.; Cicciù, M. Porphyromonas gingivalis, Periodontal and Systemic Implications: A Systematic Review. In Dentistry Journal, 2019; Vol. 7, p 114.

  1. Lines 218 to 223, is this part of the introduction?

It is not a part of the introduction, but a description to table no. 1. That is why we included it in the materials and methods in the new section 2.7, which increases (in our opinion) the readability of the table.

  1. How the samples were collected? What were the samples? Was it plaque or gingival crevicular fluid?

The collected samples were a plaque biofilm which was gently taken from the tooth surface with a loop.

  1. Information on the volunteers is missing; age, gender, smoking status…… etc

To the research were selected women and men (adults patients), non-smokers, aged 30 to 80 years for a total of 20 people. The subject of our research was not to compare the microbiome of smokers and non-smokers. First of all, we are interested in the influence of a specific diet on the development of inflammation of the periodontal tissues. Therefore, we used dental indicators to evaluate the phenomena occurring in the oral cavity. The tests were not intended to distinguish between smokers and non-smokers and are therefore not included in the table. In our research, we focused on the effects of four types of diets in blocking gum inflammation. Soft and hard bacterial plaque remaining in the oral cavity for more than 3 days contributes to the initiation of inflammation of soft tissues, which over time, after about 3 months, initiates periodontal disease, which is not indifferent to the health of the macroorganism. We took into account the hygienic dental indicators that determine the presence of soft plaque lasting more than 3 days, initiating inflammation of the soft tissues, turning into hard plaque over time. The deposition of tartar as a result of the deposition of unremoved bacterial plaque on the teeth, maintains the inflammation manifested by bleeding gums, swelling and pain. Based on the analyzed indicators, we were able to estimate the state of oral hygiene and its level.

  1. Line 176, What was the procedure? Please, explain it

We have named these procedures A, B, C and D for the purpose of identifying the techniques used more easily

  • collection of the bacterial inoculum with saliva on plates with an appropriate culture medium (non-invasive collection - without breaking the tissue continuity)- procedure A
  • performing hygienic and periodontal measurements (indicators)- procedure B
  • removal of tartar- procedure C
  • after next following 3 days, another bacterial inoculum with saliva on plates with an appropriate culture medium and sequencing (non-invasive collection - without damaging the tissue continuity)- procedure D.

  1. Do the authors mean gingival sulcus by the gingival fissure?

We meant the gingival fissure -touch the surface and transfer them further into the test tube

  1. Line 179, what was the surgery? Please, explain it

We wanted to collect material from the gingival fissure

  1. What was the growth condition of inoculated bacteria?

In order to avoid methodological repetitions, we quoted the conditions of microbial growth in our previous work by Kucia et.al. 2020

  1. What the white colonies represent in figure 3?

Bacterial colonies covered with oral yeast biofilm collected from the gingival fissure which was either  sequenced.

Results of figure 4 should be included in the results

The results in figure 4 are described in the results section in chapter 4.2. Oral Microbiota Collected from Volunteers

  1. A major part of the discussion is not related to the results and findings

Updated and moved from conclusions to discussion (amendments are marked in green). Discussion has been corrected with a new paragraph.

Reviewer 3 Report

Point 01

“The study concerned the effect of a specific diet on the induction of biofilms in periodontal diseases with the voluntary consent of the volunteers participating in the experiment, so the consent of the bioethics committee was not needed, as there was no direct invasive interference with the periodontal tissues, only a non-invasive collection of biological material for research from periodontal tissues.”

This is not entirely true. How then to classify “removal of tartar”, which was listed was one of the steps of the study protocol? This is a direct invasive interference with the periodontal tissues.

Point 02

“Whereas the parametric analysis was performed based on the Student's t-test.”

How have the authors decided by a parametric test instead of a non-parametric one?

And where are the results of such test? I was not able to find them in the manuscript. And please present the significance values.

Point 03

“The research group, when averaging the results, was at a similar level in terms of oral hygiene.”

Please present the results that allowed the authors to conclude that the patients in the research group presented “a similar level in terms of oral hygiene”.

And connected to that: is this a conclusion that was supposed to be drawn? To check the levels of oral hygiene was not even mentioned as an aim of the study.

Point 04

The conclusion is overly long, and looks more like a (poorly done) shortened version of the Results. Actually, I was able to find some new information on the Conclusion that was not mentioned in the Results section. Conclude on the main findings of the study, in a short text. And present the results properly, in the Results section.

Author Response

Thank you very much for your apt suggestions that will improve the substantive value of our manuscript. All corrections in the text are marked in green.

Point 01

“The study concerned the effect of a specific diet on the induction of biofilms in periodontal diseases with the voluntary consent of the volunteers participating in the experiment, so the consent of the bioethics committee was not needed, as there was no direct invasive interference with the periodontal tissues, only a non-invasive collection of biological material for research from periodontal tissues.” This is not entirely true. How then to classify “removal of tartar”, which was listed was one of the steps of the study protocol? This is a direct invasive interference with the periodontal tissues.

A contract was drawn up with each patient, prepared by a lawyer of the university in which the patient gave his informed and voluntary consent to the measurements (indicators) and to the voluntary, with his consent, the removal of tartar. The patient did not incur any costs. The removed calculus was not taken for the study as it was an additional service (free of charge) after biofilm removal following a specific diet where further molecular tests were performed.

Point 02

“Whereas the parametric analysis was performed based on the Student's t-test.”

How have the authors decided by a parametric test instead of a non-parametric one?

And where are the results of such test? I was not able to find them in the manuscript. And please present the significance values.

In our research, we chose the parametric test which is characterized by: greater number of assumptions to be met, greater power of tests, more accurate measurement, better interpretability of the obtained results. As a rule, less requirements must be met by the collected data in order to perform non-parametric tests, but they provide less information, according to to us they are "worth" less compared to nonparametric tests. Parametric tests - they require meeting the assumptions (although some of them can be omitted under certain conditions), but the results are more accurate and based on them, better interpretations can be made. One of the most characteristic features of parametric tests is the normal distribution of the measured variables and that the variables must be measured on a quantitative scale.

The results of the statistical significance of the student's test after applying both API and PBI indices are presented in the table 4.

Point 03

“The research group, when averaging the results, was at a similar level in terms of oral hygiene.”Please present the results that allowed the authors to conclude that the patients in the research group presented “a similar level in terms of oral hygiene”.And connected to that: is this a conclusion that was supposed to be drawn? To check the levels of oral hygiene was not even mentioned as an aim of the study.

Qualitative research is characterized by the fact that information / unexpected results appear during the research. After analyzing the indicators, it turned out (it was an unintended effect) that oral hygiene after averaging was at a similar level in all participants. Fuchsin staining and OHI and Pl.I hygiene indicators in 4 groups showed similar values ​​in terms of oral hygiene. Hard and soft plaque was found in all subjects. Because each of the studied groups of patients with plaque and tartar followed a specific type of diet. After averaging the results from each diet group, it was found that the patients had a similar oral hygiene status. Regardless of the diet used, the measurements of hygiene indicators were similar. Differences began to appear at the level of plaque build-up in the interdental spaces. The inflammation recognition index was lower in the W and T diets. Fuchsin stains tartar, soft bacterial plaque and plaque (in the form of deposits) resulting from drinking colored drinks. The next two indicators determine the presence and amount of hard and soft plaque. The first index was higher because it was associated with discoloration of tartar, soft bacterial plaque and deposits from consuming colored drinks - hygiene was worse (at the right level). The next two measurement indicators indicate the presence of tartar and soft deposits (bacterial plaque) made with a dental probe (periodontometer) and these values ​​were at a good level according to the indicator evaluation criteria. According to them, measurements, calculations and results are entered into the test evaluation criteria. Three hygiene indicators are listed here (see Table 2 and Figure 4).

Point 04

The conclusion is overly long, and looks more like a (poorly done) shortened version of the Results. Actually, I was able to find some new information on the Conclusion that was not mentioned in the Results section. Conclude on the main findings of the study, in a short text.

The conclusions was shortened and corrected. Part of the summary has been moved to the results section (marked in green). Result parts that are outside the chapter results have been included in it description for tables 1 and 2.

Reviewer 4 Report

The study deals with an interesting topic and tries to answer the following question: can different types of diet affect oral microbiology and generally oral and periodontal health?
In my opinion, the study presents the methodological limits of recruitment of participants both in terms of number and inclusion criteria, and the type of study is not even specified (case-control, cross-section? Epidemiological? Longitudinal?).

1. Table formatting: Check the formatting of the tables according to the MDPI guidelines
2. Provide more explanation in the figures
3. Arachnia propionica: I believe it is the name used in the old nomenclature, perhaps the name currently used is Propionibacterium propionicus / propionicum.
4. Check and verify the name of all bacteria that correspond to the most recent and widely used in scientific literature.
5. What were the inclusion and exclusion criteria for the participants?
6. What characteristics did the participants have (age, gender, or other?)
7. Are only 20 participants included for all 4 groups? Aren't they a bit small for a correct statistical analysis?
8. What are the limitations of this study?
9. The conclusions are in my opinion too long. I will focus on answering these 4 introductory points
1. estimation of the influence of food on causing inflammation of soft tissues in the oral cavity in the presence of residual biofilm
2. to estimate the effect of food on the supra- and subgingival biofilm, is it the same or different?
3. to analyze the types of food in terms of the possibility of slowing down the development of inflammations in the oral cavity or rejecting the above possibility
4. to estimate which products in the diet cause the greatest oxidation stress in the oral cavity of mammals

Author Response

Reviewer 4

Comments and Suggestions for Authors

The study deals with an interesting topic and tries to answer the following question: can different types of diet affect oral microbiology and generally oral and periodontal health?
In my opinion, the study presents the methodological limits of recruitment of participants both in terms of number and inclusion criteria, and the type of study is not even specified (case-control, cross-section? Epidemiological? Longitudinal?).

 Thank you very much for your apt suggestions that will improve the substantive value of our manuscript. All corrections in the text are marked in green.

  1. Table formatting: Check the formatting of the tables according to the MDPI guidelines

At this stage, we left the table formatting as we thought it would be more readable and transparent. The tables presented in the present graphical form (the so-called form) are approved by the MDPI publishing house. However, if there is a need to edit them, we will try to correct them.

  1. Provide more explanation in the figures

All figure explanation have been expanded.

  1. Arachnia propionica: I believe it is the name used in the old nomenclature, perhaps the name currently used is Propionibacterium propionicus / propionicum.

Both names function in the microbiological nomenclature. The reviewer's suggestion is presented in Table 3

  1. Check and verify the name of all bacteria that correspond to the most recent and widely used in scientific literature.

Have been checked and standardized and validated as previously used

  1. What were the inclusion and exclusion criteria for the participants?

Any adult willing to take part in our research program - could participate in it without restrictions. The exclusion consisted in the recognition and diagnosis of periodontitis in a patient over 10 years of age and discontinuation of treatment. During one visit, the chance of removing tartar in such people is rather impossible (test procedure requirement - 3 days of diet, material collection, measurements, scaling, 3 days of diet).

  1. What characteristics did the participants have (age, gender, or other?)

To the research were selected women and men (adults patients), non-smokers, aged 30 to 80 years for a total of 20 people. The subject of our research was not to compare the microbiome of smokers and non-smokers. First of all, we are interested in the influence of a specific diet on the development of inflammation of the periodontal tissues. Therefore, we used dental indicators to evaluate the phenomena occurring in the oral cavity. The tests were not intended to distinguish between smokers and non-smokers and are therefore not included in the table. In our research, we focused on the effects of four types of diets in blocking gum inflammation. Soft and hard bacterial plaque remaining in the oral cavity for more than 3 days contributes to the initiation of inflammation of soft tissues, which over time, after about 3 months, initiates periodontal disease, which is not indifferent to the health of the macroorganism. We took into account the hygienic dental indicators that determine the presence of soft plaque lasting more than 3 days, initiating inflammation of the soft tissues, turning into hard plaque over time. The deposition of tartar as a result of the deposition of unremoved bacterial plaque on the teeth, maintains the inflammation manifested by bleeding gums, swelling and pain. Based on the analyzed indicators, we were able to estimate the state of oral hygiene and its level.

  1. Are only 20 participants included for all 4 groups? Aren't they a bit small for a correct statistical analysis?

Yes, because only so many people signed up as a result of the post-semester announcement at the university about recruitment for research. We wanted to do this, however, to construct a preliminary study and develop a methodology for more people in the future.

  1. What are the limitations of this study?

The research was limited by finances, so the removal of the stone was free of charge, the visit to the dentist, dentist or peridontologist in Polish offices costs a minimum of about 150 euros once.

  1. The conclusions are in my opinion too long. I will focus on answering these 4 introductory points
  2. estimation of the influence of food on causing inflammation of soft tissues in the oral cavity in the presence of residual biofilm
    2. to estimate the effect of food on the supra- and subgingival biofilm, is it the same or different?
    3. to analyze the types of food in terms of the possibility of slowing down the development of inflammations in the oral cavity or rejecting the above possibility
    4. to estimate which products in the diet cause the greatest oxidation stress in the oral cavity of mammals

Yes, but these are not conclusions but goals

Round 2

Reviewer 2 Report

I would like to the authors for revising the manuscript. Some points still need to be addressed: 

Line 288, "Determining diet of sequencing on Pathogenic Bacteria" this sentence doesn't make sense. 

Figure 3, include in the figure legend what each plate represent. 

The microbiology assay is not correct as it shows fungal overgrowth in the culture plates. 

Why do the authors choose student t test although the study includes 4 groups to compare between?

Author Response

Reviewer 2

Thank you very much to the reviewer for valuable comments that contribute to improving the quality of work

Line 288, "Determining diet of sequencing on Pathogenic Bacteria" this
sentence doesn't make sense: the sentence has been corrected to:

3.2. Analysis of bacterial biofilms in search of periopathogens of bacterial complexes

Figure 3, include in the figure legend what each plate represent.

Figure 3 is currently Figure 1. The description below the figure has been corrected as shown below

Figure 1. Examples of bacteria on growth media after collecting the bacterial inoculum from patients marked as F,B, W,T (see chapter 2.2.),relating to each type of diet.Grown fungal colonies (including yeasts) indicate a serious oral cavity infection and insufficient hygiene, therefore only grown bacterial colonies were used for sequencing, in line with the assumptions of the study.

The microbiology assay is not correct as it shows fungal overgrowth in the culture plates.

The appearance of fungi on the plate with the growth medium is not an infection or a methodological error, but only the contents of the oral cavity of subsequent participants of the study and indicates a very poor condition of oral hygiene, which, however, was presented in the form of indicators presented in table 2 in the results section

Why do the authors choose student t test although the study includes 4 groups to compare between?

Student's t-tests are used to compare TWO groups with each other. We agree that You cannot compare several groups with each other by performing the Student's t-test several times. If we have more than 2 groups, we have to use other statistical tests. Therefore, we strongly agree with the reviewer's suggestion so we used the tukey test as described below. Thank you for drawing our attention to this.

The description below has been used in section 2.6 Materials and methods

Statistical analyses were performed using Statistica (version 12, StatSoft, Tulsa, OK,
USA). The examined characteristics in different groups are presented as mean values,
with standard deviation. Results were analyzed with one-way ANOVA.

When the F ratio was significant, Tukey test was used to
determine differences between groups. Statistical significance was set at p < 0.05.

Reviewer 3 Report

The authors have properly answered to my questions. However, there is one last thing to do: two references that were recently added, namely references numbers 11 and 11a, need to be REMOVED from the manuscript. They are not exactly directly related to the study being presented by the authors, and were very probably suggested by another reviewer to try to boost citations of own previous publications of this reviewer, in an very unethical behavior. I can even see that the first author of both references is the same. Therefore, the following two references need to be REMOVED from the manuscript, only after which I will re-consider the manuscript for publication:

11.  Fiorillo, L. We Do Not Eat Alone: Formation and Maturation of the Oral Microbiota. In Biology, 2020; Vol. 9, p 17.

11a. Fiorillo, L.; Cervino, G.; Laino, L.; D’Amico, C.; Mauceri, R.; Tozum, T.F.; Gaeta, M.; Cicciù, M. Porphyromonas gingivalis, Periodontal and Systemic Implications: A Systematic Review. In Dentistry Journal, 2019; Vol. 7, p 114.

I would like to see the manuscript again after that is done, before any decision making by my side.

Thank you.

Author Response

We would like to thank the honorable reviewer for drawing our attention to both citations in the text. We removed sensitive publications as suggested by the reviewer. Instead of items [11 and 11a] in this place we included the following items presented below as number 11 and 12, which in the manuscript are marked in azure. We kindly ask you to reconsider our manuscript.

11.Marsh PD. Dental plaque: biological significance of a biofilm and community life-style. J Clin Periodontol. 2005;32(Suppl 6):7–15. doi: 10.1111/j.1600-051X.2005.00790.x.

12.Bassler BL, Losick R. Bacterially speaking. Cell. 2006;125(2):237–246. doi: 10.1016/j.cell.2006.04.001.

Reviewer 4 Report

The manuscript has improved slightly although not all points have been addressed.
I appreciate the efforts of the authors however to improve the manuscript.

1. Format the tables according to the MDPI guidelines
2. Figures 1 and 2 (I can't understand the need to insert them, I don't think they add value to the manuscript). Furthermore Figure 1 is blurred.
3. When I asked to add Study Limits I did not mean this: "The research was limited by finances, so the removal of the stone was free of charge, the visit to the dentist, dentist or peridontologist in Polish offices costs a minimum of about 150 euros ounces "
4. I would move the following chapter by integrating it into the introduction (2. Aim of the work)

Author Response

Reviewer 4

Thank you very much for your valuable suggestions which, in our opinion, significantly contributed to the improvement of the substantive quality of our work

The manuscript has improved slightly although not all points have been addressed.
I appreciate the efforts of the authors however to improve the manuscript.

  1. Format the tables according to the MDPI guidelines

The tables in the manuscript have been prepared in accordance with the guidelines of MDPI. In order to facilitate the editing of larger tables, you can use smaller fonts, but not less than 8 pt. In our case, the font is 9 points. To create tables, we used the Microsoft Word Table option.

  1. Figures 1 and 2 (I can't understand the need to insert them, I don't think they add value to the manuscript). Furthermore Figure 1 is blurred.

We have removed figures 1 and 2 from the manuscript as suggested by the reviewer,

  1. When I asked to add Study Limits I did not mean this: "The research was limited by finances, so the removal of the stone was free of charge, the visit to the dentist, dentist or peridontologist in Polish offices costs a minimum of about 150 euros ounces "

The rule of thumb for all dental examinations is to limit the participation of a potential participant to only two examinations in a given period. At the first visit, which is after a 3-day diet, the patient comes to the office and the supragingival and gingival biofilm is collected (with a loop or with sterile toothpicks). Indicators are made, and calculus is removed above and below the gingiva. Then the patient is informed about individual oral hygiene and stays on the indicated diet for another 3 consecutive days. After another three days - he comes to the office for a second visit, the purpose of which is to collect supragingival biofilm and from the gingival fissure for examination. Then, molecular and microbiological tests of the collected material were carried out. In line with the principles of Good Dental Practice, the recruitment of clinical trial participants is carried out before its commencement, taking into account the inclusion and exclusion criteria specified in the dental examination protocol. In order for the recruited group of potential participants to be as uniform as possible, the inclusion criteria specify the prognostic factors that the future participant must have, and the exclusion criteria - the prognostic factors that the future participant may not have. The inclusion and exclusion criteria should be used to select such a group of patients that will not contain exceptional cases, and its characteristics will be relevant to the population for which the treatment is planned. In other words, the similarity of patients is important for dentists and dental hygienists so that they can objectively and reliably draw conclusions about the prognostic factors in these patients that allow them to benefit from treatment. However, at the same time, taking into account the safety of the participants of the dental examination and the unambiguous and credibility of the observation during the study, the recruitment should exclude those patients whose treatment should be stopped or whose treatment could even harm due to the occurrence of adverse events and effects. Potential participants who do not guarantee the continuity of cooperation during the study period are also excluded from the recruitment (frequent travel of the patient to a distant doctor's office for: control visits, dental and diagnostic tests).

The rules of qualification are individually selected for each dental examination and may include a number of the following attributes of a potential participant:

Age, sex, the course of treatment so far, treatment with other substances including in diet, treatment with a different dosing schedule, current results of dental and diagnostic tests, coexistence of another disease, poor general condition of the patient (> 2 according to ECOG / WHO), other diseases causing advanced failure: respiratory and / or circulatory and / or kidney and / or liver, attitude of the patient to cooperation. However, whether a given patient meets the criteria for participation in a specific clinical trial is decided by the examiner (also the researcher) who conducts the study and the patient himself by signing the informed consent form to participate in the trial.

It should be mentioned that sometimes the qualification of a patient for a dental examination using double-blind randomization may mean random assignment of the patient to a control group with placebo treatment - a preparation that is not an active substance, which both the patient and the attending physician learn about. only after the completion of the dental examination.

The criteria for limiting the number of volunteers to research resulted from several reasons:

  1. The global pandemic and lockdown caused by Sars-CoV-2 virus infections meant that there are not enough people (patients) in dental offices who could participate in this type of treatment.
  2. People who were able to recruit people in different age groups for the research had a free stone removed above and below the gingiva because, as we mentioned earlier, the dental examination itself in the office is expensive. The results obtained in the form of indicators and sequencing results are, in our opinion, a trend that should be followed based on this number of patients.
  3. Announcements about recruitment for research were presented on the websites of offices and universities of the Center for Medical and Continuing Education, however only young people applied for it.
  4. In our opinion, cash prizes would be an incentive for more people participating in the study. But according to the statute of the university and the commercial activities of offices, such a solution would not take place because it is difficult to estimate the participation and time of respondents and researchers. Therefore (see point 2) they had free tartar removed above and below the gingiva as part of the research.
  5. Little awareness of young people who are just learning that it is possible to combine dental methods and peridontological diagnostics with the methods of molecular biology and microbiological engineering and classical microbiology in order to create "new screening methods for this type of research".

  1. I would move the following chapter by integrating it into the introduction (2. Aim of the work)

The purpose of the work transferred to the introduction as suggested by the reviewer

Round 3

Reviewer 3 Report

The manuscript now seems to be suitable for publication.